# The Feedback Form and Its Role in Improving the Quality of Trauma Care

**DOI:** 10.3390/ijerph19031866

**Published:** 2022-02-07

**Authors:** Hany Bahouth, Roi Abramov, Moran Bodas, Michael Halberthal, Shaul Lin

**Affiliations:** 1Trauma and Acute Care Surgery, Division of Surgery, Rambam Health Care Campus, Haifa 3109601, Israel; h_bahouth@rambam.health.gov.il; 2The Ruth and Bruce Rappaport Faculty of Medicine, Technion, Institute of Technology, Haifa 3109601, Israel; m_halberthal@rambam.health.gov.il; 3Department of General Surgery, Rambam Health Care Campus, Haifa 3109601, Israel; ro_abramov@rambam.health.gov.il; 4The Israeli National Center for Trauma & Emergency Medicine Research, The Gertner Institute for Epidemiology and Health Policy Research, Tel Hashomer, Ramat Gan 5262100, Israel; moranb@gertner.health.gov.il; 5Department of Emergency Management and Disaster Medicine, Sackler Faculty of Medicine, Tel-Aviv University, Tel-Aviv-Yafo 6997801, Israel; 6Rambam Management, Rambam Health Care Campus, Haifa 3109601, Israel; 7Department of Endodontics and Dental Trauma, Rambam Health Care Campus, Haifa 3109601, Israel

**Keywords:** feedback, emergency medical services, trauma centers, wounds, injuries

## Abstract

*Background*: One of the tasks of a level I trauma center is quality improvement of level II and level III regional hospitals and emergency medical services by means of continuous education and learning processes. One of the tools for this, which provides constant monitoring of the quality of treatment, is feedback. The purpose of the study was to evaluate the effect of feedback on the quality of trauma care. *Methods*: Retrospective cohort study comprising two periods of time, 2012–2013 and 2017–2018. The study group included physicians and pre-hospital staff who treated patients prior to referral to the level I center. Upon arrival when the trauma teams identified issues requiring improvement, they were asked to fill in feedback forms. Data on patients treated in the trauma shock room for whom feedback forms were filled out were also extracted. *Results*: A total of 662 feedback forms were completed, showing a significant improvement (*p* ˂ 0.0001). The majority of the medical personnel who received the most negative feedback were the pre-hospital staff. A significant increase was revealed in the number of feedbacks with reference to mismanagement of backboard spinal fixation, of the pre-hospital staff, in 2012–2013 compared to 2017–2018 (*p* < 0.001). Improvement in reducing the time of treatment in the field was also revealed, from 15.2 ± 8.3 min in 2012–2013 to 13.4 ± 7.9 min in 2017–2018. *Conclusion*: The findings show that feedback improves the treatment of injured patients. Furthermore, constantly monitoring the quality of treatment provided by the trauma team is vital for improvement.

## 1. Introduction

The main causes of death among individuals up to the age of 45 years is usually due to fatal injuries from road accidents, violence, or falls [1,2,3,4]. Over the past few decades, trauma care systems have shown a decrease in mortality due to severe trauma injuries from 40% to 15% [5,6]. This decrease in mortality rate, as a result of improved trauma care, is attributed to several factors, including the development of trauma care systems that include pre-hospital management, shorter time from the place of injury to the emergency department (ER), and regional level I trauma centers that provide immediate availability of surgical teams [7,8]. One of the roles of a level I trauma center is quality improvement of level II and III regional hospitals and Emergency Medical Services (EMS) by means of continuous education and learning processes [9]. Pre-hospital trauma care by EMS staff should provide initial care and transport of the injured patient to the most suitable trauma center. High standards of care and updated protocols by the EMS increase the probability of survival and prevent life-long disabilities [10,11]. The EMS makes decisions by means of a process known as “field triage,” which includes evaluation of the physiology, anatomy, and type of injury as well as the unique considerations of the specific patient. Their goal is to transport the injured to the hospital most suitable to manage the patient’s specific injuries in a proper and timely manner, as required, based on the specific “field triage” guidelines [8,12].

Constant improvement and feedback of treatment to the personnel are vital, as explained above. Feedback is aimed to aid health professionals in improving the care they provide. Feedback has been integrated into resident education, and it has been shown to improve surgical performance [13,14]. Regular monthly feedback on inter-hospital transfer of patients with major traumas to level I centers has improved the quality of care and clinical outcomes [15]. Frequently, primary hospitals are provided negligible or even no feedback on the outcome of injured patients whose care began at their hospital. When such feedback is not provided by the receiving hospital and the staff in the trauma care unit, assessment cannot be made in order to facilitate improvements [16]. Learning from errors can generate effects of corrections [17,18]. When feedback is given with explained suggestions for corrections, the health care staff can integrate the information and improve performance [19]. 

Rambam Health Care Campus (RHCC), a level I trauma center, is a regional center that serves almost 2.5 million inhabitants and 10 level II hospitals. The RHCC emergency room annually treats 40,000 trauma cases yearly, with an average of 4000 hospitalizations per year, and 1100 severely injured with an injury severity score >16 annually. The patients arrive from five regions. Every region has an EMS headquarters and several EMS stations with level II hospitals that receive the injured in that region. Patients inflicted with severe injuries are transported directly to the level I center from all five regions by EMS or receive emergency treatment and are then transferred from the primary hospital to the level I center (RMHC). In addition to being a level I trauma center, RHCC is dedicated to providing trauma education outreach activities for quality improvement for both prevention and treatment. With the aim of improving the quality of the trauma care provided to patients in the region, RHCC designed a feedback form for EMS staff and level II and III and other regional hospitals to provide feedback on the medical treatment of patients referred to RHCC for treatment.

The feedback served as quality improvement commencing from 2012 and continuing to date. The feedback is based on two parts. The first relates to the treatment provided at the pre-hospital level, including airway management, neck collar fixation, backboard spinal fixation, pain management, namely pain assessment and treatment, and the field file. The second is the treatment provided at the primary hospital in which the feedback is focused on aspects of hospital management, including imagining, laboratory tests, emergency department files, electrocardiograms, mechanism of injuries, time of treatment, and Glasgow coma scale scores. The aim of this study was to evaluate the effect of the feedback forms used in the RHCC on the quality of trauma care provided by physicians and EMS staff in the primary treatment of trauma patients.

## 2. Materials and Methods

This is a retrospective cohort study of medical personnel and trauma patients recorded in the level I trauma center at Rambam Health Care Campus, in Haifa Israel. The study included 2 periods: 1 January 2012 to 31 December 2013 and 1 January 2017 to 31 December 2018. The study group comprised physicians from the surrounding level II and III trauma centers and pre-hospital (EMS) staff who treated patients prior to their referral to the level I, trauma shock room (TSR) during the 2 periods. When the level I trauma center team identified mistreatment of the casualties admitted, they were asked to fill in a feedback form. The feedback form (Appendix A) consisted of 2 charts, one for the EMS or pre-hospital management and the other for the level II/III and other primary hospital teams. 

The trauma records of all the patients’ data were from the trauma registration unit at RHCC. The inclusion criteria for the patients were casualties with injuries treated in the TSR with a diagnostic injury code and an abbreviated injury scale between 800 and 955.6 according to International Classification of Diseases, Ninth Edition (ICD-9). The study was approved by the RMC Helsinki Committee (0406-20-RMC)

The use of 2 time periods facilitated the assessment of the efficacy of using the feedback forms on treatment quality improvement: the feedback reports were filled out when an issue requiring improvement was noted in the remarks given during the 2012–2013 period, and the effects on treatment were reexamined during the 2017–2018 period. The feedback includes records of the response time—total time from the call until arrival at the scene and the time from arrival until departure from the scene, treatment at the scene, pain management, backboard, and cervical spine fixation, Glasgow Coma Scale (GCS) evaluation, attached field documentation, and the mode of transportation (e.g., ambulance, aerial transport). For patients who were transported from other regional hospitals, primary management, documentation, and transportation were evaluated. The feedback also included demographic data on the injured patient (e.g., age, gender), place of occurrence and diagnosis of injuries according to the Glasgow Coma Scale (GCS). Exclusion criteria included patients who were not treated in the TSR during the years 2012–2013 and 2017–2018 or who were treated in those years but did not receive feedback forms due to the fact that no corrections of treatment were noted, patients who were transported by private car, and patients who died at the scene.

The preliminary examination of the patient was carried out at the time of arrival to the emergency department (ED), vital signs upon admission to TSR were taken, and the treatment in the TSR was recorded, including in the operating room and at the final destination. According to the severity of the injury, the director of the trauma team decided whether the patient should be transferred to the TSR. The treatment was reviewed and in cases in which issues for improvement were noted, a feedback form was filled out. To ensure anonymity, all EMS personnel and the primary hospital teams (level II and III) were given a number instead of names, where the director of the trauma unit (H.B) was the only one who knew the number and names. The number of the case file was used to identify the EMS treated for the EMS headquarter. All the feedbacks (in 2012–2013 and 2017–2018) were reviewed and approved by the medical director of the trauma unit (H.B), and after approval sent confidentially to a secured email address at each primary hospital to the ED director and EMS headquarters. Phone calls were held on a weekly basis with the head of the ED and EMS headquarters to discuss the feedback. Nevertheless, the core team in the hospital did not change throughout the 2 periods (head of trauma, the senior trauma surgeons, and the trauma nurse coordinator)

### Statistical Analysis

SPSS software, version 24.0 (IBM Corporation, SPSS^®^ software Chicago, IL, USA) was used for statistical analysis. One-way Analysis of Variance (ANOVA) was used to compare continuous variables between groups. The Chi-Square Test (X2) and Fisher Exact Test were used to analyze categorical variables. A statistical significance of 0.05 (𝛼) was established for analysis. A post hoc test (Tuskey Test or Bonferroni Test) was performed to compare between the groups, the pre-hospital team, and primary (level II/III) hospitals for statistically significant differences.

## 3. Results

The results show an improvement in the quality of treatment by the EMS and pre-hospital team. A reduction in the feedback numbers show that progress in treatment was achieved. A significant decrease was revealed in the number of feedback forms filled out during the two periods, i.e., 393 forms were completed in 2012–2013 compared to 269 completed in 2017–2018, indicating the effect of feedback in improving the trauma care system (*p* ˂ 0.0001. Chi-Square).

During the years 2012–2013 and 2017–2018, a total of 662 feedback forms were completed. Of the total patients hospitalized during 2012–2013 and 2017–2018, 55% were transported by EMS teams, and 25% from level II or III hospitals, and the remaining 20% arrived at the ED by private cars. During the years 2012–2013, 6,475 patients were hospitalized, where 2013 patients were treated in the TSR. Of the 6,869 patients hospitalized during the years 2017–2018, 2403 were treated in the TSR.

The prominent causes of traumatic injuries (TI) that were treated in TSR, included motor vehicle collisions (MVCs): 198 (50.6%) in 2012–2013 and 118 (44.2%) in 2017–2018. These were followed by falls 120 (30.7%) and 87 (32.66%), respectively (see Figure 1). The trauma distribution according to age and gender is shown in Figure 2. The average age was 33.7 (±23.10) in 2012–2013 and 41.95 (±24.51) in 2017–2018, with males being most prevalent among the injured in both periods, 308 (78.6%) and 195 (72.5%), respectively. The differences between the groups of ages in males and females were constant with predominance of males with TI over females. The age group of 10–18 was represented significantly more in 2012–2013, namely, 91 (29.6%) compared to 34 (17.7%) in 2017–2018 (*p*≤ 0.001; Fisher exact test Figure 2).

The majority of constructive feedback comments were for pre-hospital EMS staff, 317 (80.66%) in 2012–2013 and 201 (76.02%) in 2017–2018 (Table 1), whereas for primary hospital teams, there were only 76 (19.34%) and 68 (24.71%) of such feedback, respectively (2012–2013 compared to 2017–2018), as shown in Table 2.

A significant increase was revealed in the number of feedback forms to EMS with remarks concerning the management of backboard spinal fixation, with 19 (6%) compared to 67 (33.5%), and neck collar fixation 22 (7%) with 73 (36.5%) received in 2012–2013 compared to 2017–2018 (*p* < 0.001, Fisher exact test). The same tendency was found with statistical significance in reference to backboard spinal fixation with 2 (3%) versus 21 (31%), and neck collar fixation, with 3 (4%) versus 27 (37%) for the hospital team (*p* < 0.001, Fisher exact test). There was an increase in comments to the EMS team regarding pain management, 46 (11.7%) compared to 59 (29.3%), respectively (*p* < 0.001, Fisher exact test). Nevertheless, the EMS showed an improvement in field files that arrived with the trauma patients from 309 (97.5%) comments in 2012–2013, to 173 (86.5%) in 2017–2018 (*p* < 0.001; Fisher exact test, see Table 2). The main improvement of field files that arrived with each trauma patient within the EMS (pre-hospital) group was in intensive care (IC) ambulance that treated and transferred trauma patients to the level I trauma center, with a reduction in feedback comments concerning missing field files from 152 (97%) in 2012–2013 to 28 (54%) in 2017–2018 (*p* < 0.001, Fisher exact test: Figure 3).

An increase in feedback comments concerning treatment documentation from primary hospitals that was missing and should have been with the patient upon arrival at the level I was revealed, with 4 (1%) in 2012–2013 versus 52 (26%) in 2017–2018 (*p* < 0.001; Fisher exact test). Nonetheless, an improvement, namely a decrease in feedback comments, were found in the comparison between 2012–2013 and 2017–2018, with 69 (91%) compared to 37 (54%), respectively, with reference to the field files arriving with the patient when transferred to a level I trauma center at the primary hospital (*p* < 0.001, Fisher exact test; Table 2). Additionally, an improvement in reducing the time of treatment in the field was revealed, with 15.2 ± 8.3 min in 2012–2013 and 13.4 ± 7.9 min in 2017–18 (*p* = 0.038). No changes were revealed in the GCS between the periods in reference to EMS staff, with an average of 9.4 ± 5.8 and 10.2 ± 5.3 (*p* = 0.10) and regarding primary hospitals with 6.5 ± 5.4 and 8.1 ± 5.8, respectively.

## 4. Discussion

The establishment of trauma centers with improved protocols and redistribution of ambulance dispatch centers have resulted in reduced response time and have succeeded in reducing the death toll among trauma injured patients [20]. Periodic courses on pre-hospital trauma life support and advanced trauma life support (ATLS) for all medical teams have also improved treatment [21,22,23]. Constant monitoring of the quality of treatment is lacking and depends on the perception of the individual in the field of EMS or the medics at primary hospitals. Failing to correct errors will consequently increase the probability of recurrence. This point of view is consistent with numerous well-established theories of learning [17,24]. The follow-up was conducted by phone between the director of the trauma unit (H.B) and the correspondent director of the ED. In addition, an annual meeting was held between EMS personnel and the directors to review the feedback and improve cooperation Furthermore, an annual convention was held in the region to learn from mistakes and promote cooperation between level I, II, and III hospitals and EMS.

The authors believe that mandating and tasking a governmental agency with oversight over data collected, as well as formulating a system under which it is possible to trace the data to facilitate quality improvement, can significantly increase the response to feedback, as well as enable better use of the data for improvement of treatment.

Feedback and corrective actions are crucial in cases of mismanagement [25,26,27]. Moreover, no benefit can be gained unless the feedback results in solutions for incorrect treatment [27,28]. In this study, data were collected from the years 2012–2013 and compared to data collected four years later for the period of 2017–2018. The aim of this study was to evaluate the effect of feedback on trauma teams in improving the quality of care. The feedback is based on two parts. The first relates to treatment at the pre-hospital level, including airway management, neck collar fixation, backboard spinal fixation, pain management, namely pain assessment and treatment, and the field file. The second is treatment in primary hospitals, in which the feedback focuses on aspects of hospital management, including imagining, laboratory tests, emergency department file, electrocardiograms, mechanism of injuries, time of treatment, and GCS.

Though the outcome of the feedback form has shown improvement in quality over the years, great efforts should continue to reduce repeated mistakes. In an attempt to understand why certain treatment was inconsistent in improving treatment, two reasons emerged. First, the turnover of the personnel of the EMS and in the training of the ED staff at the hospital has had a great impact on the QI after 4 years. Secondly, the controversy in the literature concerning treatment such as backboard and neck collar fixation complicates matters.

There were more forms filled out for EMS teams than for referring hospitals since the number of patients referred to the TSR was greater (55%) compared to those from level II and level III hospitals (25%). Moreover, in 2017–2018 the number of trauma patients at TSR were higher, (2403 patients) compared to the number of patients (2013 patients) in 2012–2013, which emphasizes the improvement and the effects of the feedback. This project, which began as a quality improvement (QI) project focused on feedback forms has been found to improve the quality of trauma team care [29,30,31]. The feedback forms filled out in 2017–2018, for backboard and neck collar fixation significantly increased compared to those received in 2012–2013. Neck collar and backboard fixations are used in order to prevent secondary injury of spinal and neck injuries during transfer or treatment of patients. Usually, motor vehicle collisions are the main cause of neck and spinal injuries, followed by falls [32]. This method has been adopted by pre-hospital medical services around the world and is advocated in trauma courses (PTHLS and ATLS) [33,34]. In recent years, however, there has been evidence of the opposite benefit in the use of neck collars and spinal fixations, which can cause severe damage due to the fixation [35,36]. Such damage includes pain, pressure sores, increased intracranial pressure, prolongation of the length of stay, difficulty in intubation, and risk of spinal fractures in adults [37,38,39,40,41,42]. These implications might explain the change in use between the periods as described. The paradox of poor spine care and the continued use of the traditional methods by EMS, reinforces the necessity of a joint protocol regarding such treatment, which should be made available to all levels of medical personnel. This could be a significant benefit of the use of feedback for significant improvement.

The most unsatisfactory outcome from the feedback is the deterioration in pain management by EMS with an increase in feedback from (*n* = 22; 7% in 2012–2013 to *n* = 73; 36.5% in 2017–2018 *p* < 0.001), indicating the need of constant feedback and meticulous and constant monitoring of treatment [43]. Consequently, this issue needs to be addressed in order to find a solution and to bring this concern to the knowledge of the EMS.

Moreover, though an improvement was found in reducing the time of treatment in the field, more efforts should be made to reduce this time even more. Aerial transportation was introduced after the initial period, which has helped improve the time of arrival to the level I trauma centers. Time is a vital factor in traumatic injuries, as quick and accurate treatment can significantly decrease morbidity and mortality [44]. An analysis of the feedback documents from primary hospitals revealed unnecessary time wasted on CT scans, which coincides with the results of an earlier study that identified unnecessary CT scans before transference of patients to the level 1 trauma center [45]. This study shows that though feedback has improved the work of EMS and the primary hospital teams, inaccuracies in treatment continue to occur, indicating the need for constant monitoring.

## 5. Conclusions

The need to constantly monitor the quality of treatment provided by the trauma team in order to improve it is vital. The findings of this study show that feedback improves the treatment of injured patients. Aerial transportation is an important tool, which has helped improve the time of arrival to the level I trauma centers. Unnecessary time wasted on CT scans from primary hospitals before transference of patients to the level 1 trauma center should be avoided. Nevertheless, because the findings also indicate that treatment mistakes recur, constant feedback is a necessity. It is important to analyze the feedback yearly; to find tendencies or repeated mistakes that need to be addressed. More research is warranted to further assess the influence of feedback on mortality and morbidity in trauma patients.

In addition, in light of the recent COVID-19 lockdowns worldwide, a study should be conducted on their implications on feedback forms as well as on the trauma care provided by physicians and EMS staff during this very unique period.

### Limitations

The conclusions of this study should be limited in light of possible confounding factors that were not taken into account during the data collection process. For example, there may have been changes to the EMS and referring hospitals’ teams’ performance due to training, educational programs, and/or changes in staff and their competencies, which also might have had an effect on the improvement in addition to the feedback forms.

## Figures and Tables

**Figure 1 ijerph-19-01866-f001:**
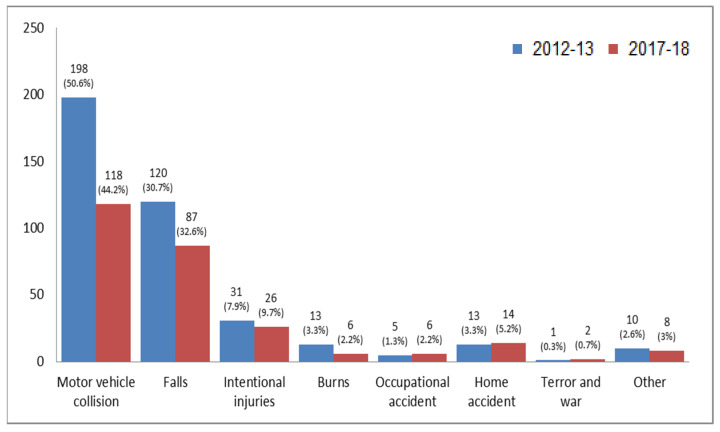
Type of trauma incident according to the years 2012–2013 and 2017–2018.

**Figure 2 ijerph-19-01866-f002:**
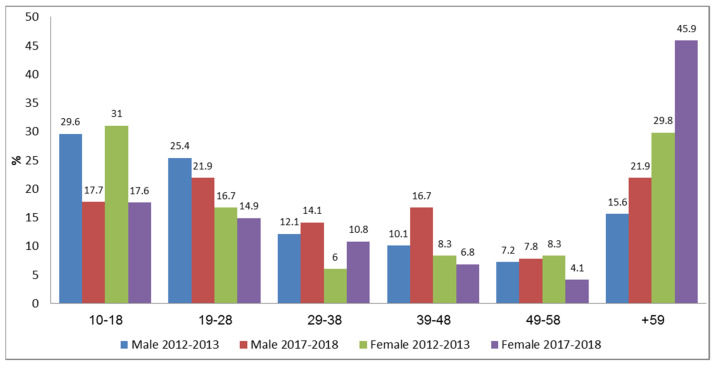
Distribution of traumatic injury according to age and gender.

**Figure 3 ijerph-19-01866-f003:**
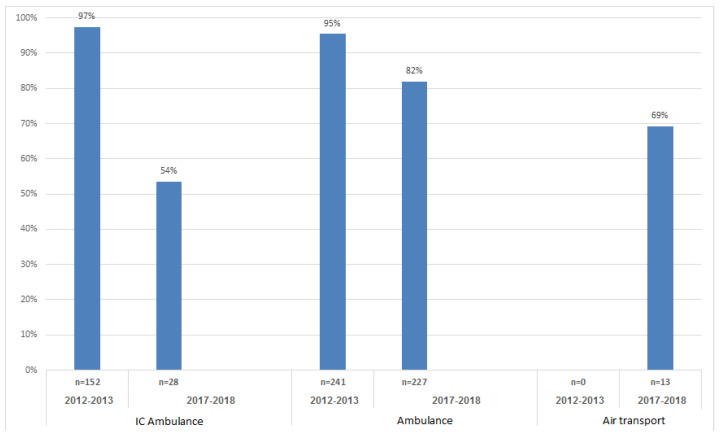
The intensive care (IC) ambulance with the main improvement in field files from the pre-hospital group arriving with the patient.

**Table 1 ijerph-19-01866-t001:** Pre-hospital team feedback.

Variables	2012–2013; *n* = 76 (%)	2017–2018; *n* = 68 (%)	*p*-Value
* Age	32.7 ± 24.9	41.9 ± 25.75	*p* < 0.001
# Male Female	55 (72) 21 (28)	49 (72) 19 (28)	*p* = 1.00
~ Mode of transportation			
Code 1 = Ambulance	2 (3)	1 (1.5)	*p* = 0.51
Code 2 = IC ambulance	74 (97)	66 (97)	
Code 4 = Air transport	0	1 (1.5)	
# Injury			
Code 1 = Motor vehicle collisions	32 (43)	23 (34)	
Code 2 = Falls	30 (41)	22 (33)	
Code 3, 4, 8 = Intentional injuries	5 (7)	9 (19.5)	
Code 5 = Burns	4 (5)	0	
Code 6 = Occupational accident	0	2 (3)	
Code 7 = home accidents	1 (1)	4 (6)	
Code 9 = Terror and war	1 (1)	1 (1.5)	
Code 10 = Other	1 (1)	2 (3)	
# Primary hospital management (see Appendix A # 4) (1 = yes comment)	9 (12)	9 (13)	*p* = 0.81
# Pain management 1 = yes comment)	7 (9)	14 (21)	*p* = 0.062
# Backboard fixation not performed (1 = yes comment)	2 (3)	21 (31)	*p* < 0.001
# Neck collar fixation not performed (1 = yes comment)	3 (4)	27 (37)	*p* < 0.001
# Pain management	5 (7)	14 (21)	*p* = 0.015
# GCS ≤ 8 GCS > 8	52 (69) 23 (31)	38 (56) 30 (44)	*p* = 0.12
GCS	6.5 ± 5.4	8.1 ± 5.8	*p* = 0.083
# Documentation from hospital (1 = yes comment)	4 (1)	52 (26)	*p* < 0.001
# No field file (1 = yes comment)	69 (91)	37 (54)	*p* < 0.001
# Surgery performed	51 (67)	40 (60)	*p* = 0.39

* *t*-test, # Fisher exact test, ~ Pearson chi square. GCS = Glasgow Coma Scale

**Table 2 ijerph-19-01866-t002:** Primary hospital team feedback.

Variables	2012–2013; *n* = 317 (%)	2017–2018; *n* = 201 (%)	*p*-Value
* Age	33.9 ± 22.7	41.9 ± 24.2	*p* < 0.001
# Male Female	253 (80) 63 (20)	146 (72.5) 55 (27.5)	*p* = 0.053
~ Mode of transportation			
Code1 = Ambulance	150 (47)	27 (13.5)	*p* < 0.001
Code 2 = IC ambulance	167 (53)	161 (80.5)	
Code 4 = Aerial transport	0	12 (6)	
# Mechanism of injury			
Code 1 = Motor vehicle collisions	166 (52)	95 (48)	
Code 2 = Falls	90 (28)	64 (32)	
Code 3,4,8 = Intentional injuries	26 (8.5)	13 (6.5)	
Code 5 = Burns	9 (3)	6 (3)	
Code 6 = Occupational accident	5 (2)	4 (2)	
Code 7 = Home accident	12 (4)	10 (5)	
Code 9 = Terror and war	0	1 (0.5)	
Code 10 = Others	9 (3)	6 (3)	
* Time in field (min)	15.2 ± 8.3	13.4 ± 7.9	*p* = 0.038
# Pre-hospital management (see Appendix A # 2) (1 = yes comment)	45 (14.5)	34 (17)	*p* = 0.45
# Pain management (1 = yes comment)	46 (14.4)	59 (29.3)	*p* < 0.001
# Backboard fixation not performed (1 = yes comment)	19 (6)	67 (33.5)	*p* < 0.001
# Neck collar fixation not performed (1 = yes comment)	22 (7)	73 (36.5)	*p* < 0.001
# GCS ≤ 8 GCS > 8	145 (46) 168 (54)	74 (37) 124 (63)	*p* = 0.054
GCS	9.4 ± 5.8	10.2 ± 5.3	*p* = 0.10
# No field file (1 = yes comment)	309 (97.5)	173(86.5)	*p* < 0.001
# Surgery performed	150 (47)	107 (54)	*p* = 0.15

* *t*-test, # Fisher exact test, ~ Pearson chi-square. GCS = Glasgow Coma Scale,

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
