# Peer review of "The Feedback Form and Its Role in Improving the Quality of Trauma Care"

_ijerph, 2022, doi:10.3390/ijerph19031866_

Round 1

Reviewer 1 Report

Thank you for the opportunity to review the manuscript “The feedback form and its role in improving the quality of trauma care”.  The manuscript covers an interesting topic however there are some major methodological concerns that has to be revised.

The aim was to evaluate the effect of feedback forms on the quality of trauma care performed by Physicians and EMS staff.

In the method section (page 2): The study include two periods, 2012-2013 & 2017-2018. To be able to understand if there are any confounders to the described result, more information in both background as well as in the method section on any other differences (not feedback) that occured between the time periods is needed. For example, has there been any changes among the staff and their competencies?

Result section: To understand your result, I suggest that this section starts with answering the aim of the study. Now, there is a great deal of result that describes differences in trauma management between the time periods. It is difficult to understand how any of these differences could have been the result from feedback when there is no information on how the feedback did reach the staff at the time of the event. 

Reviewer 2 Report

The paper deals with interesting topic about improving the quality of trauma care. The purpose of this work was to evaluate the impact of feedback on the quality of accident care. The study includes doctors and pre-hospital staff and trauma teams. The feedback forms they have completed have been processed. From this data, problems were identified that needed to be addressed to improve the situation. The study includes more than 600 feedback forms, which is a remarkable amount. All information obtained was analyzed in detail and processed into this study. The benefit is an improvement in shortening the treatment time in the field, which is a significant benefit. Negative feedback motivates staff and highlights issues that may not be visible at first glance. Findings show that feedback improves the treatment of injured patients. In addition, continuous monitoring of the quality of treatment provided by the traumatic team is vital for improvement. In this regard, I consider this study to be extremely beneficial and highly topical, especially in this complicated time marked by the significant impact of the pandemic situation on health care but also on people's daily lives. It is important to note that perhaps today the most visible importance of this work and its important significance. Technically, the article is processed at an excellent level. The methods used have been chosen appropriately and the processing is at a high level. I did not find any serious shortcomings in the article, but I still have comments that will help improve the quality of this article:

Comments:

  1. The results of the statistical analysis in Chapter 3 are given, but they are quite confusing in the text and it is necessary to present them in graphical form as graphs. If possible, I also propose to present tables 1, 2, 3, 4 in graphic form, which has a better presentation ability than tables. Some tables will need to be broken down into several graphs and will provide clearer information from this study.But let the authors consider it.
  2. Any such activity that can save lives is extremely important, and I consider it important to continue this work. In conclusion, please indicate your plans for the future as you wish to continue this work.

Decision:

The article is extremely beneficial and after processing my comments, I recommend publishing this article.

Reviewer 3 Report

Some comments are suggested:

  • In the abstract it would be interesting to include a brief description of the background.
  • It would be interesting if the keywords are DeCS/MeSH descriptors.
  • In the introduction, although the phenomenon has been described, some results obtained on the use of feedback in the care of trauma patients between levels could have been provided.
  • In the methodology it is necessary to clarify whether informed consent was requested from the participants.
  • The total EMS staff in the included hospitals and their distribution could be explained to better understand the sample.
  • The statistical tests carried out from which the p-value is obtained must be clearly specified in the results. It is only indicated for table 2 in the text. Legends must be included indicating if it is ANOVA, Chi square, etc.
  • Clarify in the tables what GCS means
  • In the discussion of lines 220 to 226 the objectives are referred to more specifically. These objectives are not so clear in the methodology or at the end of the introduction. It is advisable to explain them in those sections mentioned above, to give greater coherence to the analysis and results.
  • In the paragraph that begins on line 234, citations of similar studies should be included so that it is adequately justified and discussed.
  • In the paragraph that begins on line 254, it is important to indicate whether or not there are studies that obtain the same results.
  • The conclusions require more specificity regarding the results, making some recommendations for practice.

Round 2

Reviewer 1 Report

Thank you for the opportunity to review the manuscript “The feedback form and its role in improving the quality of trauma care”.  There are still  major methodological concerns that has to be revised.

In the method section: To be able to understand if there are any confounders to the described result, more information in both background information  as well as in the method section on any other differences (not feedback) that occured between the time periods is needed. For example, has there been any changes among the staff and their competencies, educational requirements or other continuing professional developement during the time periods?
